# Protective Effect and Mechanism of Placenta Extract on Liver

**DOI:** 10.3390/nu14235071

**Published:** 2022-11-29

**Authors:** Liu-Hong Shen, Lei Fan, Yue Zhang, Ying-Kun Zhu, Xiao-Lan Zong, Guang-Neng Peng, Sui-Zhong Cao

**Affiliations:** 1The Key Laboratory of Animal Disease and Human Health of Sichuan Province, The Medical Research Center for Cow Disease, College of Veterinary Medicine, Sichuan Agricultural University, Chengdu 611130, China; 2School of Agriculture & Food Science, University College Dublin, D04 V1W8 Dublin, Ireland

**Keywords:** placenta extract, liver injury, oxidative stress, liver inflammation, hepatocyte apoptosis, liver fibrosis

## Abstract

The placenta contains multiple biologically active substances, which exert antioxidation, anti-inflammatory, immunomodulatory, and delayed aging effects. Its extract can improve hepatic morphology and function: on the one hand, it can reduce liver interstitial collagen deposition, lipogenesis, and inflammatory cell infiltration and improve fibrosis; on the other hand, it can prevent hepatocellular degeneration by scavenging reactive oxygen species (ROS) and inhibiting inflammatory cytokine production, further improve hepatocyte apoptosis and necrosis, and promote hepatocyte regeneration, making it a promising liver-protective agent. Current research on placenta extract (PE) mainly focuses on treating a specific type of liver injury, and there are no systematic reports. Therefore, this review comprehensively summarizes the treatment reports of PE on liver injury and analyzes its mechanism of action.

## 1. Introduction

The placenta contains multiple biologically active substances, such as amino acids, trace elements, hormones, and cytokines [1,2], making it a source of natural medicine. Its extract (PE) has various biological actions, including antioxidation, anti-inflammatory, immunomodulatory, anti-apoptosis, anticancer, hair growth promotion, and antidermal aging activities (Table 1). It was recorded in the Compendium of Materia Medica (Bencao Gangmu) that “the placenta benefits qi, tonifies blood, nourishes liver and kidney”. The liver is the largest digestive gland but also an essential metabolic and immune organ in the body; it is the main site of biological oxidation and has a strong metabolic capacity. It is indispensable for maintaining life and has multiple functions, including glycogen storage, protein synthesis, hormone synthesis, and detoxification [3,4,5]. Various biotic factors (bacteria, viruses), abiotic factors (chemical drugs, physical damage), and unhealthy lifestyles (high-energy diet, binge drinking) can cause liver injury and induce chronic liver diseases [6,7,8,9]. Many chronic liver diseases gradually progress to liver fibrosis, liver cirrhosis, or hepatocellular carcinoma, ultimately leading to irreversible hepatic lesions [10,11] (Figure 1). Currently, many medicines for liver disease treatment are unable to achieve practical therapeutic effects, with chemoresistance and undesirable side effects [12,13]. Therefore, seeking a safe and effective therapeutic agent for treating liver diseases has become a hotspot in current medical research. In recent decades, PE has progressed in treating chronic hepatitis, fatty liver, liver fibrosis, and cirrhosis. This review summarizes PE’s mechanism of action and discusses some of the content of some studies. nutrients-14-05071-t001_Table 1Table 1Main functions of PE.FunctionsObjectsSubstancesAntioxidationHuman placentaPorcine placentaGoat placentaCow placentaUridine, L-tyrosine, L-phenylalanine [14], Collagen polypeptides [15], L-tryptophan [16], Extract [17]Water-soluble proteins [18], Hydrolysate [19], Extract [20]Antioxidant peptides [21]Ribonuclease inhibitor [22]ImmunomodulatoryHuman placentaGoat placentaHydrolyzate [23]Protein extract [24], Immunomodulatory peptides [25], Immunoregulatory factor [26]AnticancerHuman placentaPorcine placentaCow placenta—Mesenchymal stem cell [27]Extract [20]Lipopolysaccharide [28]Placental growth factor [29], Placenta-specific 1 [30]Hair growth promotionHuman placentaCow placentaExtract [31], Extracellular matrix hydroge [32]Extract lotion [33], Extract [34]Skincare Wound healingPorcine placentaEquine placentaHuman placentaProcine placentaExtract [20,35]Extract [36]Immunoglobulin isotype [37], Extract [38,39], Placental extract gel [40]Extract [41]Anti-inflammatoryHuman placentaPorcine placentaSheep placenta—Extract [42,43,44,45,46], Hydrolyzate [23]Hydrolysate [19], A water-soluble portion [47]Extract [48]Cryopreserved placenta extract [49]Anti-apoptosisHuman placentaSheep placentaExtract [31,50], Hydrolysate [51], JBP485 [52], Laennec [53]Extract [48]OtherPorcine placentaOvine placentaExtract: obesity treatment [54], facilitate memory and learning [55]Promote mammogenesis, lactogenesis, and galactopoiesis [56]
Figure 1The development of the normal liver toward cirrhosis or liver cancer (figure was created using BioRender (Biorender, Toronto, Canada)). Chemical drugs, physical damage, and nutritional disorders can cause the normal liver to develop cirrhosis or liver cancer via two major pathways. (1) The factors above may induce inflammatory cell infiltration, hepatocyte degeneration, reactive oxygen species (ROS) production, and oxidative stress (OS) in the liver, further activating hepatic stellate cells (HSC) and transforming them into myofibroblasts (MFBs). The activated HSC and MFBs express high alpha-smooth muscle actin (α-SMA) levels and secrete extracellular matrix (ECM), resulting in mild liver fibrosis. There is a gradual increase in fibrous tissue and pseudolobule, and the liver structure is damaged, resulting in severe liver fibrosis and, ultimately, cirrhosis or liver cancer [57,58,59]. (2) These factors cause adipose infiltration in hepatic interstitial cells and hepatocyte steatosis, resulting in simple steatosis. Then, simple hepatic steatosis develops from steatohepatitis to fatty liver fibrosis and eventually to fatty cirrhosis or liver cancer [60,61]. Liver fibrosis and fatty liver fibrosis are reversible, and the risk of cirrhosis could be significantly reduced by early intervention or treatment. PE can inhibit or reverse the above changes and play a protective or therapeutic role.
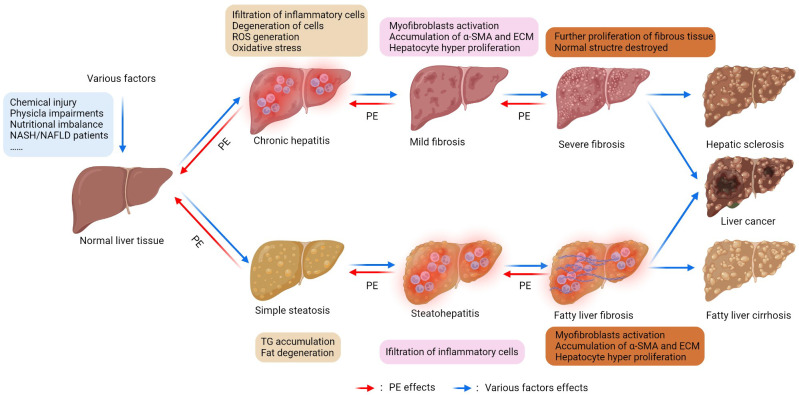


## 2. PE Improves Liver Histological Structures

The histological structures are the most intuitive indicators to evaluate the liver health status. PE has been reported to possess hepatoprotective effects against chemical factors (drugs, alcohol) [42,62,63], physical factors (partial hepatectomy) [64], and nutritional disorders (methionine and choline deficiency, high salt intake) that induce liver histological structure damage [50,65]. For example, PE could improve the vacuolar degeneration and steatosis of hepatocytes [42,51,64], inhibit the process of hepatocyte apoptosis and necrosis [51,66], and restore normal cellularity, including a typical polygonal morphology, normal cytoplasm and nucleus, and clear cell edges [62]. For the interstitial changes in the liver, PE reduces the dilatation and congestion of hepatic sinusoids and central veins and inhibits lipogenesis and inflammatory cell infiltration [62]. Further, PE attenuates the proliferation of fibrosis tissue, collagen, and fat deposition in the damaged liver [65,67] and inhibits pseudolobule formation [64], ultimately alleviating liver fibrosis and the fatty liver process [42,50,68]. In addition, a hepatoprotective agent with the placenta as the adjuvant and the transplantation of chorionic plate-derived mesenchymal stem cells (CP-MSCs) can also reduce hepatocyte necrosis, inflammatory cell infiltration, and collagen and fat deposition [63]. Transplantation of chorionic plate-derived mesenchymal stem cells (CP-MSCs) can also reduce inflammatory cell infiltration, improve liver collagen and fat deposition, and further reduce liver fibrosis and cirrhosis [67]. In summary, PE shows excellent hepatoprotection toward liver histological structures exposed to the above pathogenic factors.

## 3. PE Improves Liver Function

Liver function is canonically measured by several enzymes, including alanine aminotransferase (ALT), aspartate aminotransferase (AST), alkaline phosphatase (ALP), gamma-glutamyltransferase (GGT), and leucine aminopeptidase (LAP). In particular, lactic dehydrogenase (LDH) is a widely accepted indicator of liver injury. In addition, serum total protein (TP), liver glycogen, bile secretion, and the metabolic capacity of the liver to indocyanine green (ICG), bilirubin (BIL), fat, and alcohol can also be used to assess the severity of liver injury [69,70,71,72]. On the one hand, PE decreases the serum levels of ALT, AST, ALP, GGT, LDH [51,62,68,73,74,75,76,77,78], total bilirubin (TBIL), cholesterol (CHOL), triglyceride (TG), and non-esterified fatty acid (NEFA), and increases hepatic phospholipid and serum TP levels [62,64,66,79]. On the other hand, PE enhances the metabolic actions of the liver to bromosulfalein (BSP), iron, and alcohol, such as facilitating the scavenging of BSP and alcohol, decreasing hepatic iron deposition, and recovering the urine iron concentration [52,53,80]. Bile promotes the digestion and absorption of fat and fat-soluble vitamins secreted by hepatocytes. PE facilitates hepatic bile secretion by promoting sphincter movement and gallbladder contraction [52,53]. Moreover, PE supplementation decreases exercise-induced serum lactate elevation, increases hepatic glycogen content [74], alcohol dehydrogenase (ADH) and acetaldehyde dehydrogenase (ALDH) activity, and decreases the area under the curve (AUC) and the maximal concentration (C_max_) of alcohol [73]. However, the level of LDH increased in serum after administering PE in a report on alcohol-induced liver injury. The reason is that LDH is widely distributed in all organs, especially in the heart, liver, and muscle. After a single administration of ethanol and PE, the level of LDH in serum cannot change significantly, and the elevation of LDH might indicate liver diseases, malignancies, heart diseases, and hematological diseases [73].

In addition, a hepatoprotective agent with the placenta as an adjuvant can also reduce ALT, AST, ALP, and TBIL levels in serum [63]. After CP-MSC transplantation in the liver, the ICG metabolism increased and returned to normal levels, and TBIL levels decreased [67]. The placenta-derived stem cells (PDSCs) uptake ICG, store glucose as glycogen, and generate urea [81]. These effects are observed either by pretreatment or incubation with PE over the entire culture period, which indicates that PE could promote PDSCs’ differentiation into hepatocytes and exhibit some hepatocellular functions. In a study, the intravenous administration of PE was superior to subcutaneous administration, which was related to drug adsorption speculatively [66]. Interestingly, PE had different effects on AST in freshly isolated and primary cultured hepatocytes. The former increased while the latter decreased [82], possibly because the isolated hepatocyte membrane was damaged, and due to the additional effect of PE on the membrane; moreover, the therapeutic effect of PE is attenuated after pasteurization [68], presumably due to the inactivation of some components by high temperatures. In conclusion, PE can improve liver function by reducing liver enzyme levels and restoring the liver metabolism, synthesis, and secretion functions of BIL, fat, and alcohol.

## 4. PE Improves Liver Oxidative Stress

PE contains various antioxidant components, such as uracil, L-tyrosine, L-phenylalanine, L-tryptophan, and collagen peptides. Among them, the antioxidant activity of the mixture of uracil, L-tyrosine, and L-phenylalanine accounts for around 46% of PE, and it can effectively scavenge free radicals and decrease the malondialdehyde (MDA) level [14]. The antioxidant activity of collagen peptide accounts for around 15% of PE, and it can degrade deoxyribose and scavenge free radicals [15]. L-tryptophan exerts a potent free radical scavenging effect by inhibiting the lipid peroxidation induced by oxidative stress, and its antioxidant activity is even higher than the mixture of uracil, L-tyrosine, and L-phenylalanine mentioned above [14,16].

Multiple pathogenic factors cause an increase in ROS production, which leads to OS and hepatocellular injury. For this reason, there are a variety of antioxidant systems, especially antioxidant enzymes, in the body. The critical antioxidant systems include glutathione peroxidase (GSH-Px), superoxide dismutase (SOD), and catalase (CAT), which are said to be the body’s first line of defense against ROS [83]. Furthermore, there are particularly high concentrations of glutathione (GSH) in the liver, which buffers the redox equilibrium of the cell by undergoing oxidation or reduction reactions, according to the redox potential of the cell [84]. In addition, phase II detoxification enzymes, including heme oxygenase-1 (HO-1) and quinone oxidoreductase 1 (NQO1), are involved in detoxifying OS. HO-1 regulates the balance between free heme and bound heme to prevent the accumulation of free heme, which is essential to avert heme toxicity. Moreover, HO-1 suppresses the generation of ROS by downregulating several components of nicotinamide adenine dinucleotide phosphate (NADPH) oxidase-4 (NOX-4), including p22 phox and p67 phox [50,85]. NQO1 is known to maintain alpha-tocopherol (vitamin E) and coenzyme Q10 in their reduced antioxidant state, making it an even more potent antioxidant to protect cells against OS [86]. The above antioxidants work together to exert antioxidant effects and maintain an oxidant–antioxidant status balance in the body.

PE exerts antioxidant effects mainly through activating the Nrf2 pathway, increasing the antioxidant enzymes’ (SOD, CAT, GSH-Px) [51,62,74] and phase II detoxification enzymes’ (HO-1, NQO1) [51,65] activity, and reducing the generation of MDA and ROS in the liver [63,65,76]. Furthermore, PE reduces the expression of liver Nox4, p22phox, p67phox, and 4-hydroxy-2-nonenal (4HNE, a product of lipid peroxidation) and decreases the intracellular redox potential. Endothelial oxidative stress impairs eNOS activity and reacts with NO to form peroxynitrite, subsequently leading to a vicious cycle of endothelial injury. PE enhances the survival rate and increases the eNOS activity of liver sinusoidal endothelial cells (LSECs) [50,53,65,87]. The authors speculated that these beneficial effects could be partly attributed to the protective effect of PE on LSECs [50]. In addition, PE also increases the hepatic GSH levels and protects hepatocytes by activating the endogenous antioxidant systems of the nuclear factor erythroid-2-related factor 2 (Nrf2)/HO-1/NQO1 pathways, such as increasing the gene and protein expression of Nrf2, HO-1, and NQO1 [65]. Interestingly, several investigations have reported that the individual administration of PE decreased GSH levels compared with the blank group [62,77]. The reason might be because PE has lowered the free radical generation, and it then causes a slight decrease in GSH levels to maintain redox homeostasis in the liver. However, further verification is required. In addition, after the administration of PE, the activity of SOD decreased in a study. The reason was the drug-induced liver lipid peroxidation and ROS generation, which caused a corresponding increase in SOD activity, and PE plays a suppressive role during this process [77,88]. It can be seen that PE does not purely improve the antioxidant capacity but maintains redox homeostasis in the liver.

In vitro assays showed that PE increases the HepG2 protein expression of sequestosome-1 (SQSTM1, p62), HO-1, and Nrf2 in the nucleus while decreasing its expression in the cytoplasm. Moreover, PE decreases the expression of liver myeloperoxidase (MPO), nitric oxide (NO), and Kelch-like ECH-associated protein-1 (Keap1), which indicates that PE scavenges ROS by increasing the expression of antioxidant enzymes via the Keap1/Nrf2/p62 signaling pathway [51]. Mitochondria are the major site of ROS production; conversely, they are vulnerable to ROS attack. Pretreatment of HepG2 cells with PE restored the morphological structure of mitochondria, increased the mitochondrial mass, and elevated the mitochondrial membrane potential (ΔΨm) of hepatocytes. In conclusion, PE ameliorated mitochondrial injury and restored mitochondrial function [88]. The above studies illustrate that PE ameliorates hepatic oxidative stress and has good in vitro and in vivo antioxidant activity.

## 5. PE Improves Liver Inflammation

Inflammation is an adaptive response triggered by foreign pathogens or tissue injury and involves many complex interactions between cellular and inflammatory mediators, closely associated with OS and liver fibrosis [89]. Supplementation of PE reduces the accumulation of macrophages and decreases the concentrations of inflammatory biomarkers, such as tumor necrosis factor-alpha (TNF-α), interleukin (IL)-1β, IL-6, and IL-10 [50,51,53,62,74,75,88]. Furthermore, PE increases serum interferon-gamma (IFN-γ) [74] and immunoglobulin G2a (IgG2a) [88] levels, which suggests that PE could enhance the body’s immune function. Intercellular adhesion molecule-1 (ICAM-1) is an important adhesion molecule that mediates the adhesion reaction and leukocyte migration, usually used as an indicator of inflammation [90]. PE decreases the mRNA and protein expression of ICAM-1 and inhibits the interaction of hepatocytes with lymphocytes [75]. Moreover, immunohistochemistry revealed that ICAM-1 is expressed in the amnion and chorion of the placenta [91], suggesting that it may be involved in the inflammatory process. Several investigations have reported that PE suppresses T cell activation and proliferation, and the hepatocytes showed increased levels of anti-inflammatory factors, including IL-5, granulocyte colony-stimulating factor (G-CSF), fractalkine, IL-10, and IL-13, and decreased levels of pro-inflammatory factors such as IFN-γ, IL-1β, IL-2, IL-3, IL-12, and TNF-α. Furthermore, decreases in soluble cd40 ligand (sCD40L), FMS-like tyrosine kinase 3 ligand (Flt 3L), and granulocyte macrophage-colony stimulating factor (GM-CSF) levels were also noticed [68]. It can be seen that PE could alleviate liver inflammation. However, further studies are needed to investigate the regulatory role of PE in signaling pathways.

## 6. PE Improves Liver Apoptosis and Autophagy

Apoptosis refers to the autonomous and orderly death of cells controlled by genes to maintain a stable internal environment, which is associated with various morphological and functional changes, such as cell shrinkage, chromatin agglutination, DNA fragmentation, apoptosome formation, and the expression levels of pro- and anti-apoptotic factors. Furthermore, autophagy is a process involving the phagocytosis of cytoplasmic proteins or organelles into vesicles and fusion with lysosomes to form autophagic lysosomes, which degrade the contents of the lysosomes, thereby realizing the metabolic needs of the cells themselves and leading to the renewal of some organelles. Thus, apoptosis and autophagy can reflect the situation in cell injury [92,93,94]. Annexin V is a phospholipid-binding protein and can specifically bind to the phosphatidylserine of early apoptotic cells, which is used in the measurement of cellular apoptosis [95]. Propidium iodide (PI) is a cell-impermeable dye that only stains dead cells or late apoptotic cells with damaged membranes [96]. Therefore, the combination of these two methods can better detect apoptotic cells. Moreover, cysteinyl aspartate specific proteinase-3 (caspase-3) was demonstrated to be crucial for poly(ADP-ribose)polymerase (PARP) cleavage and DNA fragmentation, which are regarded as apoptotic hallmarks [51].

PE exerts anti-apoptotic effects by reducing the DNA breaks or laddering and the number of apoptotic hepatocytes showing Annexin V- or PI-positive (Annexin V+/PI+) values, increasing B-cell lymphoma-2(Bcl-2) expression, and decreasing Bcl-2-associated X protein (Bax) and caspase-3 expression, and inhibiting the cleavage of PARP [51,75,76]. Similarly, PE increases the expression of anti-apoptotic factors Bcl-2 and Bcl-xl in liver endothelial cells [50]. LC3, which exists in the LC3-I and LC3-II forms, is a well-accepted autophagy marker involved in autophagosome formation [97]. PE inhibits drug-induced autophagy in HepG2 cells, and decreases LC3 conversion from LC3-I to LC3-II (the autophagosome marker) and the expression of autophagy-related factors, including DRAM, CHOP, P53, ATG8, cTSd, BEcN1, LAMP1, ATF4, and ATF6 [51]. Taken together, PE improves liver apoptosis and autophagy, but the effects on autophagy need further study.

## 7. PE Improves Liver Fibrosis and Collagen Deposition

Liver fibrosis represents a transitional and reversible stage between chronic hepatitis and cirrhosis, which is more often seen clinically in viral hepatitis or alcohol or fatty liver. It is characterized by the excessive accumulation of an extracellular matrix (ECM), primarily collagen (Col), within the liver that destroys the normal liver architecture [98]. Therefore, it is necessary to alleviate or prevent the process of liver fibrosis. Chemical drugs, physical damage, and nutritional disorders cause liver-inflammatory cell infiltration, ROS generation, and hepatocellular degeneration, further activating HSCs and transforming them into MFBs [98]. The activated HSCs and MFBs secrete a large number of a-SMA and various ECM, including col-I, col-III, and col-IV, which disrupts the homeostasis of collagen synthesis and degradation, resulting in the accumulation of ECM in the liver and the formation of fibrotic scars gradually [99]. This homeostasis depends on the balance between matrix metalloproteinases (MMPs) and tissue inhibitors of metalloproteinases (TIMPs), which are the key factors in the degradation and remodeling of the ECM. TGF-β1 is a pivotal profibrogenic cytokine in liver fibrosis and an inhibitor of hepatocyte proliferation that can induce small mothers against decapentaplegic (Smad) 2/3 transcription and activate HSCs but is negatively regulated by Smad 7 [100,101,102,103]. In addition, the serum indexes of type III procollagen (PC-III), Col-IV, laminin (LN), and hyaluronidase (HA) can effectively reflect the condition of liver injury and fibrosis, and they are commonly used as clinical indicators to reflect liver fibrosis.

PE inhibits the TGF-β-induced expression of α-SMA, Col-I, Col-III, and Smad phosphorylation and activation to reduce collagen deposition and the fibrotic area, and it increases the activity and protein expression of MMP-9. In an in vitro study, PE decreased the expression of fibrosis-related genes, including actin alpha 2(Acta2), Col1a1, and TGF-β1 [50,65,68]. The hydroxyproline (Hyp) content could reflect collagen formation and indicate the amount and consistency of scar tissue [104]. A hepatoprotective agent with the placenta as the adjuvant can also reduce collagen deposition and the fibrotic area in the liver and decrease serum levels of HA, LN, PC-III, Col-IV, and Hyp. At the same time, the expression of fibrosis-related genes, including Col-I, Col-III, α-SMA, and TGF-β1, and the protein expression levels of Smad 2/3 and p-Smad 2/3 were decreased [63]. It could also be found that the gene and protein expression levels of Col-I and α-SMA decreased, and the activities of MMP-2 and MMP-9 increased, which can limit the synthesis and deposition of Col-I in the CP-MSC-transplanted liver [67]. Additionally, immunohistochemistry revealed that MMP-9, MMP-2, and MMP-12 were expressed in the amnion and chorion of the placenta, indicating that they have the potential to treat fibrosis [91]. In summary, PE can improve liver fibrosis and collagen deposition.

## 8. PE Promotes Hepatocyte Regeneration

Placental-derived stem cells have strong potential to differentiate into hepatocytes and repair damaged tissues, which exhibit normal hepatocyte function after stem cell transplantation of the liver. During the differentiation process of CP-MSCs into hepatocytes, their morphology progresses toward a typical hepatocyte-like polygonal morphology and they start to aggregate. After transplantation, they can uptake a large amount of ICG, store liver glycogen, and produce urea. Furthermore, the gene expression patterns are similar to hepatocytes, including the expression of hepatocyte-specific transcription factors (HNF1a, HNF4a) and hepatocyte markers (AFP, CK18, CK19, CXCR4, TAT, TTR, and ALB), which suggests that CP-MSCs can effectively differentiate into hepatocytes [67]. Human amnion and chorion cells represent a valuable source of progenitor cells with potential applications in a variety of cell therapy and transplantation procedures, and transplantation in neonatal swine and rats resulted in human microchimerism in the liver [91]. Human amniotic epithelial cells (hAEC) are promising transgene carriers for allogeneic cell transplantation into the liver, and they demonstrate immunoreactivity to genetic markers of liver lineage, such as human serum albumin (ALB) and α-fetoprotein (AFP), and they can synthesize and excrete ALB. After genetically modified cells containing the β-galactosidase (LacZ) gene (AxCALacZ) were integrated into the liver parenchyma, the liver contained this human-specific gene and did not generate any rejection [105]. The morphology of placenta-derived stem cells (PDSCs) is changed into a typical polygonal hepatocyte-like morphology, the expression of hepatocyte-related genes (HNF1a, HNF4a, AFP, CXCR4, ALB, and TAT) is increased, and ICG uptake, glycogen storage, and urea production are observed with the co-culture of PE and PDSCs [81].

Hepatocyte regeneration can rapidly replace the denatured, apoptotic, and necrotic hepatocytes, which is indispensable for the efficient functional recovery of the liver. The liver regeneration regulators, including IL-6, gp130, ABCG1, and ABCG2, are critical for the regeneration of hepatocytes by regulating cellular proliferation [68,106,107]. Cyclin A can be used as an indicator of the cell cycle. Ki-67 represents a proliferation-associated protein that can be detected only in the nuclei of proliferating cells, and it is also an index by which to evaluate the cell proliferation state [68,108]. PE promotes the expression of these liver regeneration regulators and increases the Ki-LI (the ratio of Ki-67-positive cells in all cells), liver mass, and liver regeneration rate. In vitro studies demonstrate that PE promotes the proliferation of different hepatocytes, such as WB-F344, T-HSC/Cl-6, and HepG2 [51,64,68,79]. However, the therapeutic effect of PE is attenuated after high-temperature heating, presumably due to the inactivation of some components by the temperature. The amount of DNA synthesis reflects the mitotic capacity of cells. Both PE and hepatocyte growth factor (HGF, a factor extracted from PE) promote the DNA synthesis of liver cells.

Similarly, it is ineffective for PE after heating. Moreover, when PE was fractionated on a heparin sepharose column, the mitogenic effect of PE was found to be located mainly in the heparin-bound fraction. Some modulators in the heparin-unbound fraction enhanced the proliferative activity of the heparin-bound fraction via a synergetic mechanism [79]. In addition, CP-MSCs could promote hepatocyte regeneration in vitro and in vivo, which was present as the number of ki-67 positive cells, the level of Ki-LI, and the expression of IL-6, gp130, cyclin A, and cyclin E increased. DNA methylation is one of the methods of gene epigenetic modification, and it plays an essential role in regulating gene expression. It is well known that promoter methylation typically represses gene expression [109]. CP-MSCs decrease the methylation levels of IL-6, gp130, and STAT3. These results suggest that the administration of CP-MSCs promotes IL-6/STAT3 signaling by decreasing the methylation of the IL-6/SATA3 promoters and thus inducing the proliferation of hepatic cells [110]. Wolf et al. [111] also confirmed that the placenta is rich in HGF and has robust mitogenic activity toward primary cultured hepatocytes. In summary, the above studies indicate that PE could be well differentiated into hepatocytes and promote hepatocyte regeneration.

## 9. Conclusions

PE is a natural medicine with great therapeutic potential for liver diseases, including hepatitis, liver fibrosis, and fatty liver. Its mechanism of action involves antioxidation, anti-inflammatory, anti-apoptosis, improving fibrosis, reducing collagen deposition, and promoting liver regeneration (Figure 2).

## Figures and Tables

**Figure 2 nutrients-14-05071-f002:**
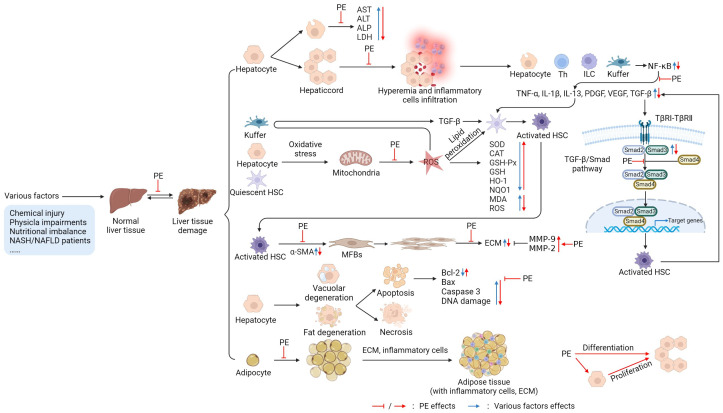
Targets of PE in protecting or treating liver (figure was created using BioRender). After liver injury, the cell membrane permeability is enhanced, or the membrane is depleted of hepatocytes; the release of liver enzymes (aspartate aminotransferase (AST), alanine aminotransferase (ALT), leucine aminopeptidase (ALP), lactic dehydrogenase (LDH)) is increased [50,51,68], and the hepatic sinus is dilated and congested due to inflammatory cell infiltration [62,63]. Damaged hepatocytes and immune cells (T helper (Th) cells, innate lymphoid cells (ILC), Kupffer cells) release inflammatory factors (tumor necrosis factor-alpha (TNF-α), interleukin (IL)-1β, IL-13, platelet-derived growth factor (PDGF), vascular endothelial growth factor (VEGF), transforming growth factor-β (TGF-β)) by activating the nuclear factor κB (NF-κB) signaling pathway, further activating hepatic stellate cells (HSCs) [58,59,112]. Injury factors can also cause oxidative stress (OS) in Kupffer cells, hepatocytes, and quiescent HSCs; increase mitochondrial reactive oxygen species (ROS) production; decrease liver antioxidant (superoxide dismutase (SOD), catalase (CAT), glutathione peroxidase (GSH-Px), glutathione (GSH)) levels; and improve malondialdehyde (MDA) levels. Moreover, ROS induces Kupffer to produce TGF-β and the lipid peroxidation of HSCs, further activating HSCs [51,77,112]. TGF-β plays a vital role in the process of activating HSCs. First, it interacts with cell membrane receptors to recruit small mothers against decapentaplegic (Smad) 2/3, forms a complex with Smad 4, and then initiates the transcription progress of the target genes by nuclear translocation and activates HSCs. Activated HSCs can release TGF-β to promote the activation of other HSCs [112,113]. In addition, activated HSCs can secrete a large amount of alpha-smooth muscle actin (α-SMA) and differentiate into myofibroblasts (MFBs). The MFBs proliferate copiously and secrete a large amount of extracellular matrix (ECM), including collagen. In contrast, matrix metalloproteinases (MMP)-9 and MMP-2 can degrade ECM [112,114]. Additionally, injury factors can also induce hepatocyte degeneration (vacuolar degeneration, steatosis) and further cause apoptosis or necrosis, which results in the levels of pro-apoptotic factors (cysteinyl aspartate specific proteinase-3 (caspase-3), Bcl-2-associated X protein (Bax)) being increased, while anti-apoptotic factors (B-cell lymphoma (Bcl)-2, Bcl-xL) are decreased and deoxyribonucleic acid (DNA) damage occurs [51,63,64]. In the interstitial part of the liver, there are adipocyte increases, adipose tissue accumulation, large amounts of ECM deposition, and inflammatory cell infiltration [67,68]. PE inhibits or reverses the above processes, plays a protective role in the liver, and promotes hepatocyte proliferation to restore the normal tissue morphology and function of the liver, as shown in Figure 2.

## Data Availability

Not applicable.

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
