# Peer review of "Protective Effect and Mechanism of Placenta Extract on Liver"

_nutrients, 2022, doi:10.3390/nu14235071_

Round 1

Reviewer 1 Report

Authors summarize literature on the protective effects of placenta extract in various pathological states of liver disease. The authors do a good job of collecting and summarizing the data as well as organizing it well. Some changes are needed however to enhance the data being presented.

Major comments:

English language editing is recommended, as there are awkwardly worded sentences. Additionally, there are numerous spelling errors throughout or missing letters from words.

Authors need to indicate throughout from section 2-8 whether human or animal models are being evaluated, the dosage and time of treatments.

Minor comments:

Line 36: it seems unlikely that long term staying up late is a cause for liver fibrosis. Authors need to provide appropriate reference for this causative effect.

Regarding figure 1, it is unclear what the forward arrow for PE indicates? Increasing disease risk?

Lines 143-145: How does HO-1 mediate NOX enzymes?

Lines 150-151: How does PE mediating antioxidant enzyme expression?

Section 10 title prospect and prospects does not make sense.

Reviewer 2 Report

This review is well-written, well-designed, and relevant to this fascinating topic; it covers the most implicated functional aspects of placental extracts in liver diseases such as hepatitis, liver fibrosis, and fatty liver, through mechanisms of action involving anti-inflammatory, antioxidant, anti-apoptosis, anticancer, immunomodulatory, skincare, hair-growth promoting, and wound healing processes. I believe it would be useful to compile a list of all current studies on this subject. I believe it is worthy of publication in this journal after minor English polishing of grammar content.

Author Response

#Reviewer 2

Dear reviewer, we are very glad to receive your comments. It's our honor to get your recognition. This review covers the most implicated functional aspects of placental extracts in liver diseases such as hepatitis, liver fibrosis, and fatty liver, through mechanisms of action involving anti-inflammatory, antioxidant, anti-apoptosis, anticancer, immunomodulatory, skincare, hair-growth promoting, and wound healing processes. We have carefully modified the grammar, vocabulary and English of the full text. Please refer to the manuscript for details.

We are very grateful for reviewer’s warm work earnestly. In all, we found the reviewer’s comments are quite helpful. They point the deficiencies about our manuscript and help us for the further improvement.

Thank you again for your comments and suggestions. I look forward to receiving your revision suggestions again as soon as possible.